# Long Non-Coding RNAs: The New Frontier into Understanding the Etiology of Alcohol Use Disorder

**DOI:** 10.3390/ncrna8040059

**Published:** 2022-08-04

**Authors:** Allie N. Denham, John Drake, Matthew Gavrilov, Zachary N. Taylor, Silviu-Alin Bacanu, Vladimir I. Vladimirov

**Affiliations:** 1Department of Psychiatry and Behavioral Sciences, Texas A&M University, Bryan, TX 77807, USA; 2Department of Psychiatry, College of Medicine, University of Arizona Phoenix, Phoenix, AZ 85004, USA; 3MSCI Program, Texas A&M University, Bryan, TX 77807, USA; 4Virginia Institute for Psychiatric and Behavioral Genetics, Virginia Commonwealth University, Richmond, VA 23219, USA; 5Departent of Psychiatry, Virginia Commonwealth University, Richmond, VA 23219, USA; 6Texas A&M Institute for Neuroscience, College Station, Texas A&M University, College Station, TX 77843, USA; 7Genetics Interdisciplinary Program, College Station, Texas A&M University, College Station, TX 77843, USA; 8Lieber Institute for Brain Development, Johns Hopkins University, Baltimore, MD 21205, USA

**Keywords:** long non-coding RNA, neuropsychiatric disorders, alcohol use disorder, postmortem brain

## Abstract

Alcohol use disorder (AUD) is a complex, chronic, debilitating condition impacting millions worldwide. Genetic, environmental, and epigenetic factors are known to contribute to the development of AUD. Long non-coding RNAs (lncRNAs) are a class of regulatory RNAs, commonly referred to as the “dark matter” of the genome, with little to no protein-coding potential. LncRNAs have been implicated in numerous processes critical for cell survival, suggesting that they play important functional roles in regulating different cell processes. LncRNAs were also shown to display higher tissue specificity than protein-coding genes and have a higher abundance in the brain and central nervous system, demonstrating a possible role in the etiology of psychiatric disorders. Indeed, genetic (e.g., genome-wide association studies (GWAS)), molecular (e.g., expression quantitative trait loci (eQTL)) and epigenetic studies from postmortem brain tissues have identified a growing list of lncRNAs associated with neuropsychiatric and substance use disorders. Given that the expression patterns of lncRNAs have been associated with widespread changes in the transcriptome, including methylation, chromatin architecture, and activation or suppression of translational activity, the regulatory nature of lncRNAs may be ubiquitous and an innate component of gene regulation. In this review, we present a synopsis of the functional impact that lncRNAs may play in the etiology of AUD. We also discuss the classifications of lncRNAs, their known functional roles, and therapeutic advancements in the field of lncRNAs to further clarify the functional relationship between lncRNAs and AUD.

## 1. The RNA World

The completion of the human genome project in 2003 demonstrated a greater need to understand the role that non-coding regions have in the classical definition of genes [1]. Approximately 80% of the human genome is biochemically active [2]. The Encyclopedia of DNA Elements (ENCODE) Consortium found that 76% of the genome’s DNA is transcribed into RNA, with only 2% of the transcribed RNA translated into functional proteins [2,3] (Figure 1). The remaining 98% are not translated and these RNA are known as non-coding RNAs (ncRNAs), also commonly referred to as “junk” or “dark matter” of the genome [4,5]. Based on their genomic organization, ncRNA can be classified into two main classes: long ncRNA and short ncRNA, with the long ncRNA further classified into regulatory or structural ncRNA [6]. Regulatory ncRNA is also delineated into small non-coding RNAs (sncRNAs) and long non-coding RNAs (lncRNAs) [6,7], which are the focus of this review. We point the reader to several excellent reviews on sncRNAs, such as microRNA (miRNA), a shorter non-coding RNA, approximately 22 nucleotides in length, [8,9] upon additional interest. LncRNAs are greater than 200 nucleotides in length, transcribed by RNA polymerase II, and have little to no protein-coding abilities due to a lack of open reading frames (ORF) [10]. LncRNAs have a similar genomic organization to the protein-coding genes, i.e., they can be 5’ capped, have exon/intron structure, and can be polyadenylated. LncRNAs can interact with other proteins involved in histone modification processes and chromatin remodeling [11]. Numerous types of lncRNAs exist with distinct functional roles in biogenesis, structure, or activity in the cell [6]. Compared to protein-coding genes, lncRNAs generally have lower expression rates; however, they display higher tissue specificity with an abundance in the brain and central nervous system, suggesting a connection to the etiology of psychiatric disorders and the emergence of behavioral phenotypes [10]. 

The first lncRNA, H19, was serendipitously discovered in the 1980s after applying differential hybridization screening of cDNA libraries of mice [12,13]. Originally classified as mRNA, H19 ultimately was classified into a novel group, non-coding RNA, due to an absence of translation [13]. A surge of discoveries in the non-coding RNA world throughout the 1990s later piqued interest in understanding the roles and abundance of the novel class.

Later studies focusing on the genome-wide classification of lncRNAs showed these to be far more abundant than previous estimates [14,15]. Research into deciphering the exact number of lncRNAs in the human genome began with the ENCODE Consortium, which aimed to continue the legacy of the Human Genome Project. Over the past decade, ENCODE rapidly increased the number of lncRNA loci and transcripts. The newest version, GENCODE 40, was recently released in April of 2022 and featured 53,029 lncRNA transcripts, a substantial increase from the 2012 version, v7, which featured 17,957 lncRNA transcripts [10,14].

## 2. lncRNA Functions

Like protein-coding genes, lncRNAs can be transcribed from either the sense or antisense strand of DNA [10]. LncRNAs exhibit a wide variety of cellular and molecular functions, depending on their subcellular localization; however, there is no universally agreed-upon classification system. Accounting for functional overlap, 11 distinct species of lncRNAs have been described, which are summarized in Figure 1. Among these, the four most discussed categories of lncRNAs in the literature are (i) long intergenic (lincRNA), (ii) antisense, (iii) sense intronic, and (iv) bidirectional lncRNAs (Figure 2) [16,17]. LincRNAs are located between protein-coding genes, whereas the antisense lncRNAs overlap one or more exons on the opposite strand of mRNA, and sense lncRNAs overlap one or more exons on the same mRNA strand. Bidirectional lncRNAs are transcribed from the promoter of a protein-coding gene in the opposite direction [17,18]. 

LncRNAs have numerous functions inside and outside the nucleus (Figure 3). In the nucleus, they participate in chromatin architecture and remodeling (e.g., chromatin organization and stabilization) [19], transcriptional regulation of protein-coding genes [20], facilitation of protein–protein interaction [20], and function as a sponge through competitive endogenous RNAs (ceRNAs) [21]. CeRNAs, also known as endogenous sponges, use microRNA response elements (MREs) to crosstalk between miRNAs and lncRNAs to act as a decoy to influence gene expression levels. Approximately 54% of lncRNAs are detected in the cytoplasm [22]. MREs target lncRNA transcripts to post-transcriptionally regulate gene expression via transcript degradation or translation inhibition. Other lncRNAs have been featured in numerous pathological functions such as mRNA stabilization, sequestration of MREs aided by miRNA, stimulation of apoptosis, and assisting in alternative splicing (Figure 3) [23,24,25]. LncRNAs are also shown to be essential to chromatin architecture. For example, the X-inactive-specific transcript (Xist), a well-known lncRNA, stabilizes the silenced X-chromosome during early embryonic development [26]. Xist, as well as other lncRNAs such as Airn and Kcnq1ot1, create a cascade of events causing Polycomb repression complexes (PRCs) to modify chromatin structure over broad regions of the genome [27]. An antisense lncRNA, HOTAIR, binds to PRC2 to promote histone H3 and lysine 27 methylation along with lysine 4 demethylation, which contributes to pathogenesis of various neurological diseases, such as major depressive disorder or cancers, such as breast cancer [28,29,30].

Considering their biological functions, not surprisingly, many lncRNAs have been implicated in multiple facets of human health and disease, such as cancer biology, diabetes, and fatty liver disease [31,32,33,34,35]. Due to their high expression in the brain, lncRNAs were also implicated in the neuropathology of psychiatric disorders. For example, two lncRNAs, GOMAFU and NEAT2/MALAT1, have been known to affect mRNA splicing of genes previously shown to be associated with schizophrenia [36]. GOMAFU affects the splicesome by directly binding to splicing factor 1 for disrupted-in-schizophrenia-1 (DISC1) and v-erb-a-erythroblastic leukemia viral oncogene homolog 4 (ERBB4), two genes that play roles in the splicesome and were shown to contribute to schizophrenia [37]. Upon dysregulation, GOMAFU interferes with the expression of splice factor proteins, leading to alternative splicing patterns to genes such as DISC1 and ERBB4 [36,37]. GOMAFU is highly enriched in neurons and facilitates neuronal regulation and support through interacting with serine/arginine-rich splicing factors during growth and development. GOMAFU also promotes neuronal differentiation during development [38]. NEAT1, PCAT1, RMRP, and NKILA are lncRNAs associated with neuropsychiatric phenotypes such as major depressive disorder (MDD), addictive behavior, and substance abuse [39,40]. NEAT1 was shown to act as a regulatory lncRNA that reduces the expression of HTT, thereby contributing to Huntington’s disease, and once again shows the broad implications that lncRNAs have on human health and diseases related to the brain [41,42].

## 3. Epidemiology of AUD

Alcohol use disorder (AUD) is a debilitating condition impacting millions of individuals worldwide. In the United States, the average 12-month and lifetime prevalence of AUD is 13.9% and 29.1%, respectively, with males historically having a higher prevalence (i.e., 17.6% and 36.0%) than females (i.e., 10.4% and 22.7%) [43]. An annual average of 87,798 alcohol-attributable deaths (AAD) and 2.5 million years of potential life lost (YPLL) occurred from 2006 through 2010, with similar numbers still occurring over a decade later [44].

The transition from DSM-IV to DSM-5 led to the merging of alcohol addiction and alcohol dependence into a single AUD diagnosis [45]. A person meeting any 2 of the 11 DSM-5 criteria during the same 12-month period would receive a diagnosis of AUD [46], with the severity of AUD—mild, moderate, or severe—based on the number of criteria met. Examples of criteria include craving for alcohol, neglect of day-to-day activities, difficulty in cutting down on drinking, and failure to fulfill major obligations due to excessive drinking (Table 1). However, integrating these separate diagnoses into one disorder, AUD, poses a challenge to future studies that utilize data obtained prior to DSM-5 implementation. Therefore, it is vital to confirm the new AUD diagnosis through access to patient records. Caution must also be taken to ensure AUD is not used interchangeably with alcohol abuse and alcohol dependency. Some of the studies mentioned below (e.g., Van Booven et al. 2021) disclose whether the patients’ diagnosis is based on DSM-IV or DSM-5 to ensure consistency between different studies utilizing DSM-IV or DSM-5 diagnostic criteria [47]. Future metastudies may also consider merging multiple datasets that represent both DSM-4 and DSM-5. AUD is also highly comorbid, i.e., it has been associated with other substance abuse disorders, major depressive disorder (MDD), and anxiety disorders [43]. Approximately 1 in 5 patients with lifetime AUD will be treated, indicating a strong need to understand genetic and environmental factors that contribute to predispositions for acquiring AUD. 

## 4. Metabolism of AUD

Alcohol is first metabolized to acetaldehyde by alcohol dehydrogenase (ADH), which is then metabolized by aldehyde dehydrogenase (ALDH) into acetate, which is utilized as an energy source in oxidative phosphorylation. Acetaldehyde is toxic when accumulated in high doses, and it can lead to tachycardia, nausea, and other unwanted effects [48]. Chronic alcohol use affects long-term symptoms such as memory loss [49] and increases the risk of developing alcohol cardiomyopathy and cancer [50]. Population-based studies have highlighted genetic variations that impact ADH and ALDH metabolic rates, causing different rates of ethanol metabolism across ethnic groups, with the ADH and ALDH metabolic rates being the lowest in the Asian populations [48]. Corollary chronic AUD can cause irreversible changes to gene expression in the brain [51]. The sedative effects of alcohol are due to the opposing action of increased activity of γ-aminobutyric acid, an inhibitory neurotransmitter [52], and decreased activity of the glutamate N-methyl-D-aspartate (NDMA) receptor, the primary excitatory receptor in the brain [52]. Alcohol releases dopamine, serotonin, and B-endorphins that contribute to relapse or cravings for alcohol [53]. The numerous molecular changes that occur from alcohol exposure highlight the critical role that alcohol has on the alteration of neuroplasticity and gene expression in the human brain. 

## 5. Neuropathology of AUD

Physiological and biochemical pathways, such as the dopaminergic nervous system of the human brain, provide ways to understand the occurrence of addiction. The mesocorticolimbic dopaminergic system comprised of the nucleus accumbens (NAc), the prefrontal cortex (PFC), and the ventral tegmental area (VTA) is a key component of the reward pathway system in the mammalian brain [53]. NAc is activated by dopamine and controls the reward system alongside the ventral striatum (VS) [52]. Upon activation, both regions have been shown to aid in the increase of cravings or dependence on alcohol [54]. A functional magnetic resonance imaging (fMRI) and resting-state electroencephalography (EEG) study performed on patients with alcohol addiction showed high levels of signal response to alcohol-related cues compared to healthy control patients in both the NAc and VS [55]. The prefrontal cortex, arguably one of the most studied regions for addiction, is an area crucial for decision-making [56]. Prone to damage from alcohol, the PFC has been hypothesized to influence craving for alcohol due to the role the region plays on dopamine release downstream in the striatum, which subsequently causes craving seen in clinical studies and postmortem analyses [57]. 

## 6. Genetics of AUD

Genetic epidemiological reports based on familial and twin studies estimate the heritability of AUD at approximately 50% [58]. However, AUD is a highly heterogeneous group of clinical and physiological symptoms (also referred to as endophenotypes) that contribute to the diagnosis. Thus, to better understand the genetic factors underlying the etiology of AUD, many GWAS are performed using AUD endophenotypes rather than the clinically heterogeneous definition of AUD. An endophenotype is a heritable and measurable component associated with an illness and must be found at a higher rate in affected individuals or relatives than in the average population [59,60]. Endophenotypes can also reflect genetically influenced pathways and clinical signs (e.g., behavioral symptoms) of a disease [59]. GWAS have identified genetic polymorphisms (i.e., single nucleotide polymorphisms (SNPs)) associated with various alcohol measures at a genome-wide significance threshold (*p* ≤ 5 × 10^−8^), such as response to alcohol, failure of impulse control, and vulnerability to impaired neurobehavioral responses to stress [59,61,62,63]. While GWAS identify the potential genomic loci contributing to the disease phenotype, they suffer from several limitations [16,64,65], the most notable being the inability to provide a functional context for the associated signals. A powerful approach to mitigate this limitation is integrating GWAS and gene expression data derived from postmortem brain expression data [66,67,68]. 

The most consistent findings from the GWAS of AUD have implicated variants residing in the alcohol metabolizing enzymes ADH and ALDH. The alcohol-metabolizing enzymes are composed of isoforms with varying functions influenced by genetic variations [48,69]. For example, ALDH2, located in the mitochondria, is responsible for the bulk of acetaldehyde breakdown into acetate [52]. Studies on different allele mutations of ALDH2 in various ethnic populations have been insightful to determine the risk of developing AUD. For example, the Asian populations are among the ethnic cohorts with the lowest risk of developing alcohol dependence, due to a homozygous mutation leading to a substitution of lysine for glutamate at position 504 that causes two defective ALDH2 alleles in individuals of Asian descent [69,70]. The homozygous recessive mutation causes the conversion of acetaldehyde to acetate to be slowed, creating an accumulation of acetaldehyde, leading to prolonged adverse effects, which deters drinking. In contrast, the homozygous carriers for the dominant and functional ALDH2 allele are predominantly present in the Caucasian and African-American descendants, leading these two populations to experience the highest number of subjects with alcohol addiction [48,69]. 

As previously defined by DSM-IV, GWAS findings for alcohol dependence have reported associations between genetic variants (i.e., ADH and ALDH subunits) and alcohol-related phenotypes in a range of populations [63,71]. However, as most genome-wide significant polymorphisms are annotated outside the protein-coding genes, a potential biological mechanism leading to developing AUD, among others, is that such SNPs can affect the function of other genomic elements such as lncRNAs [4,64,71]. Indeed, recent studies have reported polymorphisms localized within or nearby lncRNA loci associated with AUD on chromosome 4 [72]. As an example, Gelernter and team (2014) performed GWAS on alcohol dependence cases from European-American (EA) and African-American (AA) populations [24]. The findings from the two populations were then integrated and meta-analyzed, revealing approximately 50 genome-wide significant variants on chromosome 4p [63]. Among these, while the most significant polymorphism (rs1229984) in the EA cohort of the study was localized in the ADH1B locus, it appears that the association signal extends beyond ADH1B to also include LOC100507053 (a lncRNA) in the European-Americans population [63]. Subsequent conditional analyses in the EA and AA cohorts to determine the evidence for multiple independent risk loci revealed that the top significant SNP rs1229984 remains significant after adjustment for the other top variants in LOC100507053 in EA. However, this was not observed in AA, where the top rs2066702 was no longer significant after adjustment for the top SNPs in LOC100507053. This suggest that while in EA the most likely contributor of the association signal is coming from the ADH1B cluster, in the AA population, there are two independent association signals from the ADH1B cluster and LOC100507053 [63]. LOC100507053, an antisense lncRNA, is located in the same genomic region as ADH1B and therefore shares the same locus [73]. For reference, the spatial orientations of antisense lncRNAs are outlined in Figure 2.

Another meta-GWAS performed by Adkins and colleagues (2015) identified genome-wide significant polymorphisms in association with protein-coding genes (e.g., COL6A3, KLF12, and RYR3) and a singular lncRNA molecule, LOC339975. In the LOC339975 locus, only one variant, rs11726136, reached a genome-wide significance threshold [71]. In an attempt to investigate the potential functional role LOC339975 plays in the neuropathology of AUD, the research team further stratified the expression of LOC339975 based on clinical status and genotype for the top genetic variant, rs11726136, in two brain regions (i.e., nucleus accumbens (NAc) and dorsolateral prefrontal cortex (PFC)) obtained from patients diagnosed with alcohol dependence [71]. After adjusting for the confounding effects of postmortem covariates, no significant LOC339975 expression differences between cases and controls in the two brain regions were detected. However, LOC339975 expression was significantly reduced in NAc when stratified based on the rs11726136 genotypes in subjects with alcohol dependence [71]. Interestingly, LOC339975 is classified as an enhancer lncRNA on chromosome 4, which contains a region of different genes that encode ADH [63,71]. Based on these initial data, lncRNAs and chromosome 4 are undoubtedly implicated in the etiology of AUD, but the functions and mechanisms of lncRNAs in relation to influencing gene expression are still misunderstood. 

## 7. lncRNAs Functions in AUD

Behavioral measures and clinical phenotypes are essential for the diagnosis of AUD and other psychiatric disorders. Research into the underlying genetic mechanisms that contribute to AUD is still a growing field, with lncRNAs emerging as an important molecular link between the clinical phenotypes of AUD and GWAS findings. This observation has been further supported by assessing the genome-wide expression of lncRNAs in postmortem brain studies from patients with AUD (Figure 4) [47,74]. For example, in one such study, Kryger et al. (2012) found that NEAT2 is differentially expressed in the hippocampus, brainstem, and cerebellum of postmortem human brain of AUD patients [74]. Van Booven et al. (2021) also replicated the above findings, in which they showed 4 lncRNAs to be differentially expressed in the superior frontal cortex (SFC), 2 in the NAc, 15 in the basolateral amygdala (BLA), and 8 in the central nucleus of the amygdala (CNA) between 30 patients diagnosed with AUD and 30 controls [47]. 

The integration of genotypic and gene expression data can identify expression quantitative trait loci (eQTLs) that modulate the expression of key AUD genes [75]. Using a custom-build microarray approach, Drake et al. (2021) not only assessed the genome-wide expression of approximately 30,000 lncRNAs in NAc from 40 subjects with alcohol dependence and 40 controls, but by using weighted gene co-expression network analysis (WGCNA), they detected lncRNA networks associated with AUD [75]. Furthermore, by integrating their lncRNA expression and genotype data from the same sample, they identified eQTLs affecting the expression of lncRNAs in NAc from patients with alcohol dependence. They further identified and selected genes with key functions in supporting the network integrity (i.e., hubs) and tested these to determine whether their expression is under the control of specific eQTLs. Finally, the authors also identified significant enrichment of GWAS signals for other addiction phenotypes, such as smoking initiation and drinks per week, among the significant eQTLs identified in their study. 

Only a few animal model studies have been performed to implicate lncRNAs in the neuropathology of AUD, mostly due to the fact that over one-third of lncRNAs are estimated to be primate or human-specific [10]. Research on other substance abuse disorders, such as cocaine, heroin, and opioid abuse, using animal models or human postmortem brains has provided additional insight into the biological mechanisms underpinning the potential role lncRNAs play in the biology of addiction. A small cohort of four cocaine-conditioned mice versus four controls has identified 603 differential lncRNAs in the NAc [78]. Similarly, GOMAFU, NEAT1, NEAT2, and MEG3, previously implicated in AUD, were also shown to be upregulated in human postmortem brains from patients with opioid and cocaine use disorders [79,80]. One functional role of lncRNAs observed in animal models for AUD is the ability of lncRNAs to interact with proteins [11]. In an animal model for AUD, Yoon and colleagues (2012) showed that lncRNA-P21 binds with HuR, which then forms a lncRNA–protein complex that binds to let-7/Argo2 [81]. These interactions cause the stability of lncRNA-P21 to decrease and let-7 levels to increase, which are positively correlated with severity of AUD [81].

## 8. Epigenetics of AUD

Excessive alcohol drinking has also been linked to a complex interplay between epigenetic modifications and lncRNAs. Epigenetics refers to changes in gene expression and nuclear inheritance not based on changes in the DNA sequence [82]. Changes in gene expression occur through modifications and/or alterations that include changes in DNA methylation patterns, chemical modifications (i.e., chromatin remodeling) and imprinting [83]. For example, the genomic region of H19, a lncRNA found in male gametes and involved in genomic imprinting, contains differentially methylated regulatory regions (DMRs); however, the DMRs were shown to be actively demethylated when overexpression of H19 occurred in males suffering from excessive alcohol consumption [76]. Zillich and colleagues (2021) were able to identify differentially methylated regions between AUD cases and controls in the caudate nucleus (CN), VS, and anterior cingulate cortex (ACC) [84]. A very well documented example of how epigenetic regulation mediated by lncRNAs can impact addiction phenotypes is via the brain-derived neurotrophic factor (BDNF). BDNF is a critical gene involved in the survival, protection, and differentiation of neurons [77]. BDNF initiates downstream signaling through phospholipase C (PLC), phosphoinositol 3 kinase (PI3K), and Ras pathways by first binding with tyrosine-associated kinase B (TrKB) [52]. The initial contact with TrKB begins cascade events that impact cAMP-response element binding protein (CREB) expression [77]. The antisense lncRNA, BDNF-AS, overlaps the BDNF gene and acts as a negative regulator of exon IX [77]. In 2019, Bohnsack and colleagues showed that alcohol consumption at an early age increases the lncRNA, BDNF-AS, via reduced RNA methylation, which creates a cascade event in the BDNF signaling pathways within brain regions such as the amygdala [77]. Disruption of BDNF expression through lncRNA intervention during critical brain development increases the risk for AUD [77]. 

## 9. Complexity of Comorbidities Associated with AUD

Conditions in which several clinical diagnoses converge in a single person are referred to as comorbidities and can be observed at the clinical level. One example is seen in patients self-medicating themselves by drinking, which in turn causes the development of alcohol addiction in addition to major depressive disorder [43,85,86]. Comorbidities have also been shown to increase the risk of developing AUD in one’s lifetime [87]. A recent study found a strong association between AUD and addiction to smoking in patients with existing type 2 diabetes. In blood serum taken from Type 2 diabetes patients, increased expression in three lncRNAs, nuclear enriched abundant transcript 1 (NEAT1), metastasis-associated lung carcinoma transcript 1 (MALAT1/NEAT2), and NF-kappaB interacting lncRNA (NKILA), was observed [39]. NEAT2 is also documented to be overexpressed compared to human controls in the cerebellum, hippocampus, and brain stem of alcoholics without type 2 diabetes, further demonstrating the large role lncRNAs play in the neural pathways that contribute to neuropsychiatric illnesses (Figure 4) [74].

Multiple lncRNAs have been described to hold mechanistic roles in alcohol-induced organ pathologies, such as liver fibrosis and fatty liver disease. HOTAIR, a lncRNA, contains a binding site for miR-29b and has been suggested to sponge the miRNA and cause downregulation [88]. This downregulation consequentially creates methylations and further downregulation of phosphatase and tensin homolog (PTEN), which has been shown to contribute to the progression of liver fibrosis [88]. In mice, a lncRNA known as LFAR1 has been shown to bind with SMAD2/3, two proteins involved in liver fibrosis, indicating a potential role that lncRNAs have in signaling pathways that promote liver fibrosis, such as the TGFβ/Notch signaling pathway [89]. Additionally, NEAT1 performs a similar mechanism in which the lncRNA acts as a sponge to miR-146a-5p and increases steatosis, also known as a fatty liver disease, associated with AUD [90].

## 10. Therapeutic Methods Utilizing lncRNAs: A New Frontier

To date, no lncRNA-based therapeutic methods have been implemented at the clinical level for psychiatric disorders [91]. However, antisense oligonucleotides (ASOs) have provided a direct way to influence gene expression. ASOs are single-stranded oligonucleotide sequences containing between 8–50 bases [92]. ASOs bind in an antisense (complementary) manner to a particular target that induces transcriptional termination or RNA modification. Transcriptional termination occurs from the ASO-recruiting enzymes that cleave the RNA strand, such as ribonuclease H (RNase H) [91]. 

Currently, 13 ASO therapeutic treatments approved by the FDA use various mechanisms to correct a single affected gene [93]. Numerous RNA therapies are currently in phase II or III clinical trials, some targeting lncRNAs [91,94]. A recent clinical trial (ClinicalTrials.gov, NCT04259281), now in Phase II, has used an ASO-based therapy to treat Angelman syndrome, an imprinting disorder that impacts neurodevelopment and behavior in young children. In approximately 70% of cases, Angelman syndrome is caused by a deletion of the maternal copy of the ubiquitin-protein ligase E3A (UBE3A) gene [95]. The paternal allele is imprinted by the UBE3A antisense transcript, UBE3A-AS, and the latest clinical trials have used ASOs to reverse the imprinting of the paternal copy by inhibiting UBE3A-AS transcription. This is the first instance of a clinical trial utilizing a lncRNA as the molecular target [95].

A drawback currently seen in clinical trials utilizing ASOs is the requirement for detailed knowledge of the genomic region [91]. Recently, two ASO candidates failed Phase II and III of their respective drug trials for Huntington’s disease (HD) due to low efficacy or worsened outcomes due to the ASOs missing the essential target regions [41]. Similar mechanistic roles of ASOs have been utilized in pharmaceutical research on Parkinson’s disease, neuromuscular diseases such as Duchenne muscular dystrophy (DMD), spinal muscular atrophy (SMA), and even Alzheimer’s disease (AD) with EBF3-AS, an antisense lncRNA that promotes neuron apoptosis in Alzheimer’s model mice [91,92,96]. High tissue specificity and low expression levels make lncRNAs excellent candidates for a therapeutic approach that offers quicker effects at a lower dose for known targets [97]. Identifying lncRNA networks and target regions for ASOs to bind to will be critical to expand therapeutic approaches for psychiatric illnesses such as AUD, and in the next decades, the scientific community may see the first therapy utilizing lncRNAs for psychiatric treatment at the clinical level.

## 11. Conclusions

Long-term alcohol exposure contributes to the expression of altered lncRNAs through epigenetic modification and altered gene expression, thus leading to irreversible neuropathological changes. Growing evidence has continuously shown the transcriptional impacts of lncRNAs, adding an interesting element to the central dogma of molecular biology. We have discussed the cellular and molecular mechanisms by which lncRNAs could ultimately contribute to the development of neuropsychiatric disorders such as AUD. Here, we have further reviewed a growing list of lncRNAs, such as NEAT1, NEAT2, GOMAFU, BDNF-AS, and H19, contributing to AUD by summarizing the available genetic, molecular, and epigenetic studies related to the functions of lncRNAs in the etiology of AUD. 

Postmortem brain studies have identified risk lncRNAs whose expression is associated with AUD. Research through the integration of genetic and lncRNAs expression data in postmortem brain tissues from patients with AUD and neurotypical controls have identified numerous eQTLs impacting the expression of lncRNAs in the brain. Further advances in the technology and computational methods have also provided insight into the network organization of the lncRNA transcriptome, which in time will elucidate novel functional roles that lncRNAs have in the neuropathology of AUD and other neuropsychiatric disorders. A further understanding of lncRNA functionality and classifications (i.e., defining clear mechanisms of antisense, bidirectional, competitive endogenous lncRNAs, among others) will improve treatment by highlighting the genomic regions of importance, which can then be potential targets of therapeutic agents (i.e., ASOs). The research in the lncRNA field has just begun to scratch the surface of how important lncRNAs are for the normal development of the human body and mind.

## Figures and Tables

**Figure 1 ncrna-08-00059-f001:**
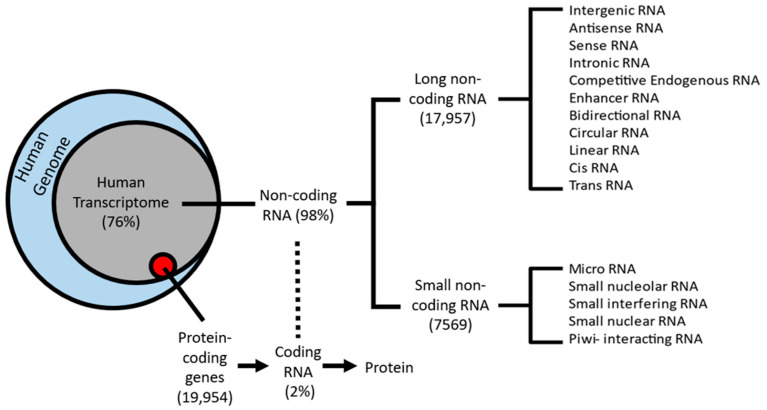
Schematic classification of the human transcriptome: 76% of protein-coding genes from the human genome (blue circle) are transcribed into RNA (gray circle), whereas only 2% (red circle) are translated into functional proteins. The remaining 98% are known as non-coding RNA. Two sub-classes emerge from non-coding RNA: small non-coding RNA (sncRNA) and long non-coding RNAs (lncRNAs).

**Figure 2 ncrna-08-00059-f002:**
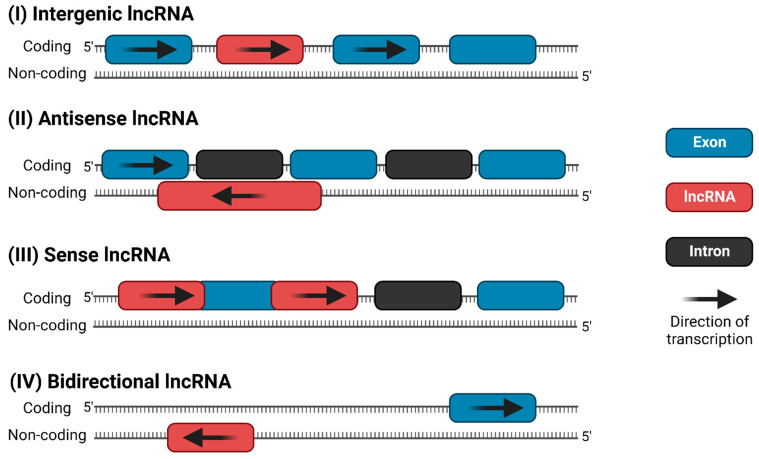
The four most common categories of lncRNAs: (**I**) long intergenic ncRNA (lincRNA), (**II**) antisense lncRNA, (**III**) sense intronic ncRNA, and (**IV**) bidirectional lncRNA.

**Figure 3 ncrna-08-00059-f003:**
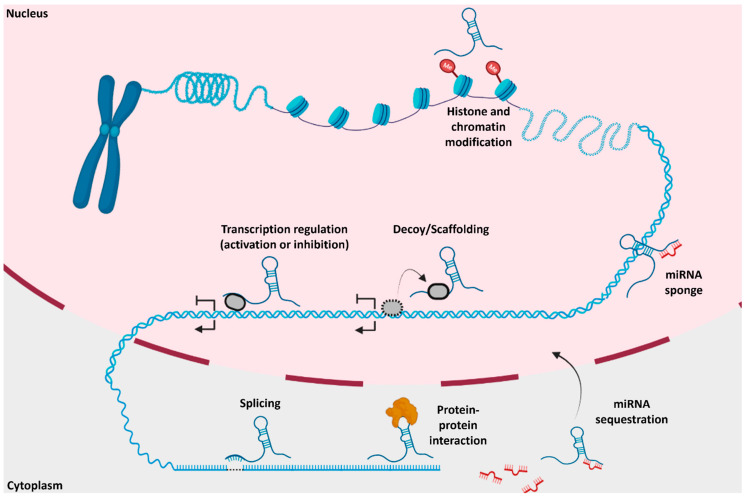
Molecular and pathological mechanisms of lncRNAs in and outside the nucleus: Various functions of lncRNAs occur, such as histone modification, transcriptional regulation, scaffolding, microRNA sequestration, splicing, and protein–protein interaction. Adapted from Hu et al. 2018 and Yang et al. 2021. Created using BioRender.

**Figure 4 ncrna-08-00059-f004:**
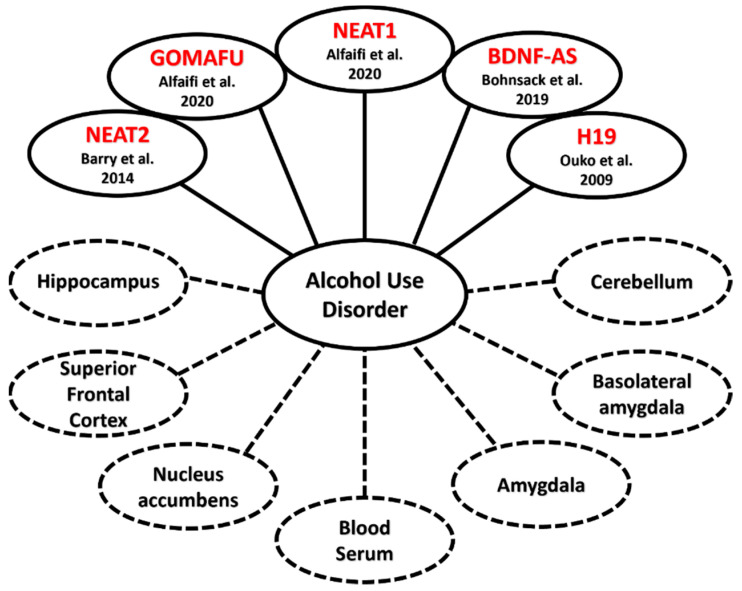
Linkages between lncRNA and AUD: lncRNAs contributing to the emergence of Alcohol Use Disorder (solid lines) and the diverse areas of the human body that samples were taken from (dotted lines), such as regions of the brain or blood serum [36,39,47,53,74,75,76,77].

**Table 1 ncrna-08-00059-t001:** DSM-5 symptoms for Alcohol Use Disorder (AUD. The presence of 2 or more of these 11 listed symptoms indicates Alcohol Use Disorder (AUD) [45]. Levels of severity are classified as Mild: 2–3 symptoms, Moderate: 4–5 symptoms, and Severe: 6 or more symptoms. Modified from the DSM-5 questionnaire given to patients [45].

DSM-5 Alcohol Use Disorder Symptoms	
1	Drinking longer or more than intended
2	Tried to quit or decrease levels of drinking, but failed
3	Sick from the aftereffects
4	Incapability to not think about drinking
5	Drinking interferes with your daily life (job, family, school, etc…)
6	Continued drinking habits regardless of daily life struggles
7	Loss of pleasure in things you once loved
8	Reckless behavior (driving, fighting, unsafe sex, etc…)
9	Continued drinking even if depressed or anxious or experiencing memory problems
10	Increased tolerance to alcohol
11	Withdrawal symptoms (shakiness, nausea, sweating, etc…)

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
