# Peer review of "Long Non-Coding RNAs: The New Frontier into Understanding the Etiology of Alcohol Use Disorder"

_ncrna, 2022, doi:10.3390/ncrna8040059_

Round 1

Reviewer 1 Report

The manuscript by Denham et al. (ncrna-1840096) “Long non-coding RNAs: the new frontier into understanding the etiology of alcohol use disorder” carefully reviews the by far available information on the field, which includes some works from the authors. I could not find many publications on this topic, and just some previous reviews published in 2017 and the present year (reference 31 of the current manuscript: https://doi.org/10.1016/j.alcohol.2017.01.004; reference 52: https://doi.org/10.1007/s00221-022-06305-x), although less focused on this specific type of ncRNA than the present work. Consequently, I consider that this Review could contribute to the upgrade of knowledge for MDPI’s readers about both, long ncRNAs and psychiatric disorders derived from alcohol use.

I have very few comments, which I hope will help the authors to come up with an even better manuscript:

Could the authors add titles that summarize the content to all figure captions? The captions have immediately been described in all figures following the number of the figure but without any title.

Specific comments:

·       Line 24: Genome-wide association studies? Please, indicate the whole name of acronyms and abbreviations the first time used in the manuscript.

·       Line 38: Please, explain that sentence further. Additionally, notice that the reference jumped from 1 to 27 in the first sentence of the manuscript.

·       Lines 11-150: please, describe further in Table 1 in the main text

·       Line 163: I miss in Table 1 the reference to the study of the questionnaire

·       Lines 179-183: a reference is needed in this paragraph

·       Line 259: Figure 4 was not referred in the main text

·       Lines 274-276: a reference is needed in this sentence

Author Response

We would like to extend a sincere appreciation to Reviewer 1. Your comments have allowed us to create a cleaner and more thorough review paper. Below are our responses.

Review 1

  1. Line 24: Genome-wide association studies? Please, indicate the whole name of acronyms and abbreviations the first time used in the manuscript.

We have corrected this error and have now indicated the whole name for GWAS.

  1. Line 38: Please, explain that sentence further. Additionally, notice that the reference jumped from 1 to 27 in the first sentence of the manuscript.

We have corrected “27” to reference “2”. We also have changed line 38 to fall more in line with the general theme of the review paper, along with a change of reference to a newer, more relevant one.

  1. Lines 11-150: please, describe further in Table 1 in the main text

We have further described the criteria listed in Table 1 within the main text

  1. Line 163: I miss in Table 1 the reference to the study of the questionnaire

We have now included a reference to Table 1 “[45]”.

  1. Lines 179-183: a reference is needed in this paragraph

We have now included a reference “[53]”.

  1. Line 259: Figure 4 was not referred in the main text

Figure 4 has now been referred in the main text at lines 291 and 364.

  1. Lines 274-276: a reference is needed in this sentence

We have now included references “[63,71]”.

Reviewer 2 Report

Generally, this review give a elucidation regarding the basic information regarding LncRNA. Then, the authors briefly demonstrated the possible involvement of LncRNA in AUD. This reviewer suggests several modifications.

1. One of my major concern is that the authors spend a large space on the basic information of LncRNA as well as the biology of AUD. Regarding the lncRNAs Functions in AUD, not enough information was provided. I suggest that if the authors would give a more thorough elaboration regarding this issue?

2. Since numerous studies investigated alcohol-related organ damage, expecially neurosystem. I wonder if this is related to AUD and could the authors add a discussion about the possible involvement of LncRNA in this and its mechanisms?

Author Response

We would like to extend a sincere appreciation to Reviewer 2. We have expanded on your comments and have addressed them in our manuscript. Below are our responses.

Reviewer 2

  1. One of my major concern is that the authors spend a large space on the basic information of LncRNA as well as the biology of AUD. Regarding the lncRNAs Functions in AUD, not enough information was provided. I suggest that if the authors would give a more thorough elaboration regarding this issue?

We appreciate the reviewer’s comment, however, we would like to point out that the lncRNA field is still nascent and virtually, with the exception of a limited studies performed by others and us, little is known about the biological mechanisms underpinning their role in the biology of addiction. Nevertheless, we have provided additional information about the potential biological mechanism of lncRNA in addiction, such as cocaine, heroin, and opioid abuse along with the use of animal model studies on said addictions. Lines 311-324.    

  1. Since numerous studies investigated alcohol-related organ damage, expecially neurosystem. I wonder if this is related to AUD and could the authors add a discussion about the possible involvement of LncRNA in this and its mechanisms?

We appreciate the review’s comment and have provided additional information regarding the role of lncRNA in organ pathology. We would like to point out further that we are among the first, if not the first, to address the involvement of lncRNA in the neuropathology of AUD (Vornholt et al. 2019 and Drake et al. 2020). Considering the liver is the main organ that metabolizes alcohol, we have provided additional details of the lncRNA role in the development of alcohol liver disorder and alcohol fatty liver. Lines 365-375.

Round 2

Reviewer 2 Report

My concerns were well addressed.